# Examining Psychosocial and Economic Barriers to Green Space Access for Racialised Individuals and Families: A Narrative Literature Review of the Evidence to Date

**DOI:** 10.3390/ijerph20010745

**Published:** 2022-12-31

**Authors:** Tila Robinson, Noelle Robertson, Ffion Curtis, Natalie Darko, Ceri R. Jones

**Affiliations:** 1Department of Psychology and Vison Sciences, University of Leicester, Leicester LE1 7RH, UK; 2Centre for Ethnic Health, University of Leicester, Leicester LE1 7RH, UK; 3School of Sociology, University of Leicester, Leicester LE1 7RH, UK

**Keywords:** psychosocial, economic, racialised, ethnic minority, green space, equality, accessibility, barriers, social prescribing

## Abstract

Background: Social prescribing (such as green social prescribing), aims to address health disparities cross-culturally to improve well-being. However, evidence highlights racial disparities in relation to access to quality green space (including local/national parks and recreational spaces). This review aimed to identify the psycho-socioeconomic barriers to green space access for racialised individuals/families and Black Indigenous People of Colour (BIPOC), to understand what cultural adaptations might be made to help support them to access green social prescribing within the UK. Method: A narrative systematic review was conducted to identify barriers to green space access for racialised individuals/families and BIPOC. Searches of publication databases (APA PsycInfo, Cochrane Database of Systematic Reviews [CDSR], Cochrane Central Register of Controlled Trials [CENTRAL], Cumulated Index to Nursing and Allied Health Literature [CINAHL], and SCOPUS Preview) were undertaken from January to February 2022, to identify quantitative peer reviewed studies. Of the 4493 abstracts identified, ten studies met the inclusion criteria and were included for final review. Results: The results suggest that interpersonal, practical (such as transportation costs, entrance fees and lodging costs) and environmental factors can act as barriers to green space access for racialised individuals/families. Most frequently reported barriers were perceptions of safety and costs associated with travel and accessing green spaces, particularly for families. Conclusion: Factors such as diversity-friendly schemes (e.g., multiple languages on signs and additional prayer spaces in parks), funding and strategies to improve safety should be considered in the design and commissioning of green space and green social prescribing initiatives in primary care. By mitigating these barriers green space can become more accessible and improve inclusivity for racialised individuals/families. Future research could explore the inter-racial differences between racialised populations and which mechanisms reduce barriers to access and in what contexts.

## 1. Introduction

### 1.1. Deprivation in Racialised Individuals/Families’

Race is a social construct, created by societies to categorise people based on their skin colour/race [1]. Terms previously used to categorise non-Caucasian/non-white people, from majority white countries have included, “person of colour”, “racial minority” or Black and Minority Ethnic groups (BAME) For this review, “Black” is used as a term to define Black-British, African, Caribbean, and African American populations. This is because the studies identified within this review are from both the United Kingdom and United States and some studies only define such populations as “Black” and do not specify the origin. [2]. These terms have been acknowledged as generalising ethnic minority/racialised groups, lacking specificity, and enforcing labels upon ethnic diversity [3], and have been superseded by terms considered more acceptable such as “racialised person”/“racialised group” [4]. It is important to acknowledge that people from the United States of America (USA) prefer terms such as Black, Indigenous and People of Color (BIPOC) to account for the erasure of darker-skinned black people and Native Americans [5]. The term BIPOC acknowledges the presence of indigenous people in the USA, that often get disregarded [6] and thus reduces further minoritisation [7]. Therefore, considering this, for the purpose of this review, non-White people from the UK will be referred to as racialised individuals/families and non-White people from the USA will be referred to as BIPOC, when referencing US studies specifically. The term we will be adopting is racialised individuals and families, unless otherwise stated.

Recent evidence suggests that people from racialised groups are at greater risk of health inequalities, which has been highlighted recently through their elevated risk of contracting COVID-19 in the current global pandemic [8], experiencing more acute symptoms, more complications and worse health outcomes [9], with higher mortality risk in Bangladeshi and Pakistani groups [10]. Migrants from racialised groups are particularly vulnerable and are more at risk of developing mental health difficulties [11]. These individuals appear less likely to access healthcare services and mental health support [12], reporting more scepticism regarding appropriateness of services and increased drop out (if they do engage), compared to White patients [13,14]. Prominent barriers to accessibility for racialised groups appear to be related to a lack of culturally sensitive provision and services need to change ‘conventional practice’ for racialised groups, however, it is apparent there is no clear strategy to tackle this [15]. Thus, burgeoning evidence argues for systemic level action, to engage racialised communities, to mitigate barriers to accessing interventions and enhance their health and wellbeing, through individually tailored, culturally appropriate care.

### 1.2. Social Prescribing

Within the UK, social prescribing connects patients to community support via social prescribers or link workers and can be one of many interventions used to improve individuals’ health and well-being. Social prescribing is a holistic approach that is used alongside usual treatment [16]. Currently there is a national roll out of social prescribers/link workers in the UK National Health Service (NHS) primary care practices through the recently formed Primary Care Networks (PCNs). Social prescribing is seen as a key mechanism to increase engagement with patients, including racialised groups, to address psychosocial issues [17]. Reviews have highlighted the benefits of social prescribing in primary care contexts by improving wellbeing, reducing anxiety/depression levels, reducing isolation, and promoting health behaviours [18,19]. Social prescribing can also decrease the burden on the NHS and has successfully reduced the number of GP appointments and prescriptions within the UK [20].

One facet of social prescribing involves referring patients to green space and activities in the form of Nature Based Interventions (NBIs). Collectively, this is described as ‘green social prescribing’ [21], and incorporates many activities (i.e., green exercise, local walking schemes, care farming, community gardening, food-growing projects, conservation volunteering, outdoor arts, and cultural activities) [22]. A £4 million, green social prescribing pilot scheme has been introduced as part of the COVID-19 recovery plan recognising that psychosocial wellbeing can be enhanced through engagement with nature. Recent reviews have articulated these benefits, revealing that engagement with NBIs is associated with improved fruit/vegetable intake, physical activity, and reduced body mass index (BMI) and can improve long term conditions [23]. Further reviews have highlighted the physiological benefits of NBIs on healthier blood glucose levels, reduced blood pressure, as well as improved mental health [24,25]. A key mechanism for engagement in green social prescribing appears to be the equitable accessibility of green spaces.

### 1.3. Green Space

Good quality green space refers to having a decent level of “accessibility, maintenance, perceived safety, presence of amenities [and an] absence of litter” [26]. Access to good quality green space appears to be beneficial for both children and families. Increased access to green space during pregnancy is associated with increased birth weight and a lower risk of obesity and neurodevelopmental difficulties in children [27].

Previous literature reviews have also shown that having access to green space, despite relative socio-economic barriers, can benefit mental health [28], reduce health inequalities [29] and improve health outcomes [30]. Yet, socio-economically deprived communities within the UK are particularly disadvantaged and have less access to good quality green spaces than their richer counterparts [31]. People from racialised groups (e.g., African/Caribbean, South Asian) are twice as likely than White individuals to live in areas of deprivation [32].

Despite the high proportion of racialised groups in socioeconomically deprived areas, recent examination of published literature has focused on barriers to green space access for socioeconomically deprived communities yet rather less on the barriers of racialised individuals/families specifically [33,34]. Even within the highly socioeconomically deprived neighbourhoods, individuals/families from racialised backgrounds seem likely to have differing reasons for accessing green spaces, with differing and/or greater barriers to access, than their White counterparts.

### 1.4. Racial Disparities in Green Space Access and Green Social Prescribing

Nature England [35] has recently shown that only 26% of Black ethnic groups spend time in the countryside, compared to 44% of White individuals. Qualitative research from the US also highlights the inequalities in access to green space via “nature engagement” for young BIPOC. Practical barriers (notably accessibility issues) were highlighted but potential psychosocial factors were also extrapolated, including fear by young BIPOC to engage in accessing green spaces and experiences of social exclusion when BIPOC do access the green spaces [36]. This study was qualitative and explored only the experiences of a small purposeful sample of the USA BIPOC population and it is important to note that there are very few studies that explore psychosocial barriers to green space accessibility for young BIPOC.

Similarly within the UK, Black and Asian people visit natural settings 60% less frequently than White individuals [37], partially explained through practical barriers (transport issues, funding cuts, safety concerns) when accessing green social prescribing, and constraining involvement with NBIs [38]. These systemic barriers are also often referred to as structural racial inequalities that are exacerbated by structural racism, notably wider political and social disadvantages within society that are experienced by racialised minority groups.

Current studies tend to focus on green social prescribing outcomes and do not explain why the natural, non-prescribed access to green space is lower for racialised communities [37,38]. Understanding quantitative evidence of the processes underpinning access to green space, using larger samples of racialised populations may inform green social prescribing schemes to develop, target and implement strategies to help engage racialised individuals and families in green social prescribing referrals and NBIs. This, particularly within UK contexts, can support delivery of green social prescribing through the National UK link worker programme in primary care [39].

### 1.5. Theoretical Underpinning for Racial Disparities in Green Space

Racial disparities in green space access may be attributed to numerous factors, including prejudice/discrimination [40], unequal care access [41] and a lack of cultural adaptation [42]. One prominent model advanced to explain this disparity is the Environmental Justice Framework (EJF) [43]. The EJF embraces the principle that all communities are entitled to equal environmental, health, employment, housing, transportation, and civil rights law/regulations affecting quality of life. The framework recognises that systems may have underlying assumptions that contribute to differential exposure and unequal access for certain groups and aims to develop strategies to mitigate disparities [43].

The EJF also suggests that certain racialised groups are indeed environmentally disadvantaged in respect of access to green spaces. Supported by geolocation research, [44,45] the framework reveals that individuals residing in socioeconomically deprived communities have less access to good quality green space than those residing in higher socioeconomic communities [31,46,47]. Yet, whilst this research maps a particular population(s) at specific time points and examines associations between factors it cannot articulate the beliefs, perceptions, and appraisals of those from racialised groups offering nuanced explanations of green space access [23].

### 1.6. International Evidence

Reduced access to green space for racialised populations, is not UK-specific but is a global issue evidenced internationally. To date, research has captured mostly homogenous communities (where most of the population are the same ethnic background), such as in China [48,49] and most majority White, affluent societies including the UK [38], Germany [50], the USA, [51] and Australia [52]. International evidence further supports the EJF and suggests that urban planning and structural development schemes disproportionately affect racialised communities as they are less likely to have access to good quality parks [53].

### 1.7. Gentrification

In line with the EJF, gentrification can act as a barrier to green spaces for racialised individuals and families. Local governments (particularly in the USA) develop low-income areas with high BIPOC populations and improve the associated green spaces. However, evidence has shown that such gentrification can further marginalise BIPOC individuals and families, as they are usually unable to afford the newly developed areas [54,55,56,57]. USA evidence has also highlighted that BIPOC have significantly less access to the green spaces geospatially but also commented on qualitative psychosocial factors.; The subjective qualities of these green spaces can draw people in, but when sight is blocked (through bushes and darkness), people can feel unsafe and disengage from accessing green spaces [58]. Additionally, urban planning in the USA has left certain marginalised communities with reduced economic investment and at greater risk of global warming impacts, with these communities suffering hotter climates than non-marginalised communities [59]. Indeed, one USA literature review has highlighted that marginalised communities experience a lower sense of community, feel more out of place in newly developed green spaces, and often use the green space less than newcomers [60].

By comparison, the racialised population in the UK is diverse, and lives in more mixed areas with varying attitudes towards perceived green space barriers [61]. Evidence to date focuses mainly on geographical barriers for marginalised communities without focusing specifically on racialised communities [60], or focuses on racialized communities without privileging the psycho-socioeconomic factors that may explain the lack of access within the UK. Understanding the psycho-socioeconomic barriers to green space access for racialized individuals and families that have been captured quantitatively will extend the evidence base of the EJF beyond US applications, and enrich understanding of how services can make green social prescribing more equitable for all.

### 1.8. Rationale

Previous literature reviews have explored barriers to green space accessibility; some have highlighted the perceived barriers from socio-economically deprived individuals [33,34], whilst others have highlighted systemic barriers reducing racialised groups’ access to green spaces [60,61]. Whilst noting that race and socioeconomic status may intersect, this review seeks to explore quantitative studies reporting on barriers to green space accessibility, specifically for racialised (particularly African, Caribbean, Hispanic and west/south-east-Asian) individuals and families within the UK and USA.

### 1.9. Review Aims

This review thus aims to:

Identify the psychosocial and economic barriers to green space access for racialised individuals and families within UK and USA contexts, to help to inform how green social prescribing interventions can be adapted to support more equitable access.

## 2. Materials and Methods

### 2.1. Search Strategy

Informed by the Cochrane Handbook [62] and guided by PRISMA guidelines for reporting [63], reviewer one undertook an initial scoping search of the Cochrane Library, Scopus, and Google Scholar to identify existing review evidence on the barriers to green space access for racialised communities.

### 2.2. Developing Search Terms

The PICO framework [64] guided the operationalisation of the search terms in this review (Table 1).

Key search terms included synonyms and terms around; individuals/families from racialised/minoritised groups, green space/outdoor recreation, and accessibility (Table A1, Table A2, Table A3 and Table A4). A specialist librarian was consulted to identify searchable terms, and search terms were identified via previous literature and reviews within the field of green space, NBIs, and racialised communities [65]. International terms were also included to widen the search for the purpose of this review.

### 2.3. Data Sources and Selection

Bibliographic databases (APA PsycInfo, Cochrane Database of Systematic Reviews [CDSR], Cochrane Central Register of Controlled Trials [CENTRAL], Cumulated Index to Nursing and Allied Health Literature [CINAHL], and SCOPUS Preview) were searched for the period 2012 to 2022.

Database searching was supplemented with a search of grey literature within the same time-period to mitigate potential publication bias [66,67]. A manual hand search of the references from relevant, similar literature [68], previous literature reviews and UK government documents [8] was also undertaken (Table A1, Table A2, Table A3 and Table A4). A second reviewer (C.R.J) reviewed the final studies for inclusion.

### 2.4. Eligibility Criteria

In relation to our review aims, the following inclusion/exclusion (Table 2) criteria were used to refine the search.

### 2.5. Screening

Records (4493) were obtained from the publication databases, and four further records were retrieved from hand-searching, before being transferred to Endnote 20 Software where duplicates were removed (N = 1011). Identified titles and abstracts of every record were screened with 3444 ineligible records removed, leaving 34 full text articles further scrutinised for eligibility. Ten records were ultimately deemed eligible for final review and the PRISMA diagram (Figure 1) [73] details the search results, screening process and reasons for exclusion at each stage.

### 2.6. Data Extraction and Synthesiscu

#### 2.6.1. Data Extraction

A comprehensive data extraction tool (using Microsoft Excel 2022) was developed and piloted (Table 3). Elements of the table were structured with the PICO framework and aspects of standardised data extraction tools [74].

#### 2.6.2. Quality Appraisal

Quality appraisal of literature was undertaken using two standardised checklists for quantitative studies. Downes’ (2016) [75] quality appraisal tool was used for seven studies given it is specifically developed for cross-sectional studies. Colour and numerical codes were applied, to appraise the quality of each study. Each item was awarded either green/‘2’ (for ‘yes’); orange/‘1’, (for ‘no’); or yellow/‘0’, (for ‘don’t know’). Where an item was not applicable (N/A), it was removed from the final total.

The three remaining studies were appraised by Kmet et al., (2004)’s [76] Standard Quality Assessment Criteria as this is used for randomised control trials. Kmet et al., (2004)’s [76] tool, coded items as yes/‘2’, partial/‘1’ or no/‘0’. A percentage based on ratings for all items was awarded in both tools. Higher scores indicated a greater study quality.

Despite the high scores obtained across the studies, the quality appraisal scores did not determine the inclusion or exclusion of studies for this review.

#### 2.6.3. Data Synthesis

A narrative synthesis approach was undertaken to draw conclusions from highly heterogenous data [77,78]. Data is grouped based on population and intervention characteristics and presented textually, in diagrams and tables. A broad comparison of study characteristics and outcomes were organised to identify patterns within the literature. It is important to highlight that due to limited number of studies that met the inclusion criteria, the narrative results were in parts generated from only one study.

## 3. Results

The ten eligible papers were all published in peer-reviewed journals between 2015 to 2020.

### 3.1. Study Characteristics (Table 4)

#### 3.1.1. Study Settings

The reviewed studies yielded samples ranging from 78 to 7506. All included studies were located in urban communities, reporting varying degrees of socio-economic status and household incomes. Most studies targeted communities with high levels of racial diversity. Eight studies were conducted in the USA; two studies were UK-based.

#### 3.1.2. Study Designs

Seven studies reported solely quantitative data and three studies used a mixed methods design (with only the quantitative results reported in this review). Two studies comprised randomised controlled trials; Lee et al., (2015) [79] who compared park use between participants who increased physical activity and participants who increased vegetable intake, and Razani et al., (2020) [68] who compared park use for people who were prescribed three organised group outings in parks, to people who were not.

The remaining eight studies utilised an observational/analytic approach, seven deploying a cross-sectional design, and one study used a cohort study design [80].

#### 3.1.3. Participant Characteristics

Studies used stratified [81], quota [65], probability [82] and momentary-time [83] sampling methods. Others used non-probability sampling approaches, such as: opportunity [84,85], volunteer [68,79] and judgment [80,86] methods.

Some studies sampled across the lifespan, reporting on adults, adolescents, and children [84,85], whilst others only included children and their mothers [68,79], or participants over the age of 16 [78] or 18 [79,81,82,83]. Most studies included male (n = 3175) and female (n = 3753) participants, however, some reported female data only [68,79,81].

The racial/ethnic groups reported in studies also varied. The two UK studies [65,80] included mainly South Asian (Indian, Pakistani, and Bangladeshi) and White ethnicities. Whereas the (remaining) USA studies, included Hispanic/Latino, Black and White ethnicities, which is understandable given the significantly larger numbers of Hispanic and Latino people living in the USA (19.8 million/18.4% [87]) compared to the UK (113,000/0.17%) [88]. Additionally, the cultural differences in the interpretation of Black people, depended on the country of study. Most USA studies characterised Black individuals, without clarifying their original diaspora, however, UK studies categorised Black into two cultural groups (African and/or Caribbean).

#### 3.1.4. Green Space Categorisation

Green space was categorised as predominantly parks/recreational spaces across all studies. One study simply used the nebulous term “green space” with no further definition [65]. Activity in green space varied across studies with some focusing on physical activity (e.g., exercise, walking. biking) [68,79,82,83,84,85,86] and play for children [80]. Others detailed activity in green spaces more precisely and in addition to physical participation included passive activities (e.g., sitting/resting/relaxing, barbequing), consumptive activities (e.g., fishing and hunting), appreciative activities (e.g., camping, bird watching) and mechanised outdoor recreation (e.g., ATV riding and motorcycling/off-road biking), [81].

#### 3.1.5. Outcome Measures

Included studies examined diverse variables that captured barriers for racialised groups when accessing green space, and by different means (details in Table 4). Some studies specified observable dependent variables, such as physical activity, park use, acculturation, access to physical activity resources, quality of physical activity resources, BMI, proximity, and satisfaction with green space [68,74,79,81,82,83,84,85].

Varying measures were utilized to assess these variables notably; observation tools (e.g., the Parks and Play Spaces Direct Observation Tool; [84]) audit tools (e.g., The Parks and Play Spaces Environmental Audit Tool; [84]), calculated energy expenditure [84], standardised scales [81] (e.g., acculturation scale for Hispanics—BAS [77]) and composite [79]/vegetation indices [80].

Most studies used self-report measures (e.g., questionnaires/surveys) and semi-structured interviews to collect participant data, and some created bespoke questionnaires for park use [68,82,83], demographics (all studies), social interaction [83], appraisals (about health, social environment, neighbourhood environment and local green space), [65], frequency of national park visits, barriers to visiting national parks and the likelihood of visiting national parks more [86]. Whilst it was unclear whether bespoke questionnaires were created based on previously validated measures, other studies used established questionnaires, such as the Strengths and Difficulties Questionnaire (SDQ) [80] (with robust validity [89]), to screen emotions and behaviours in children and young people.

A range of barriers to green space access for racialised groups were highlighted in the reviewed literature. However, it is important to note that one study [80], did not explicitly examine barriers to green space, rather the implications of the lack of green space for South Asian populations. Salient aspects of this data were thus extrapolated and synthesised into the themes reported.

**Table 4 ijerph-20-00745-t004:** Study characteristics.

Publication Details (Author/Publication Year)	Location	Study Aims	Participants	Racialised Communities	Intervention Study Design Variables	Outcome Measures	Outcomes/Results	Key Findings
Das et al., (2016) [82]	Minneapolis, MN (USA)	Increase awareness of park related health benefits and remove specific park use barriers among minority and foreign-born communities.	Probability Sampling method. Sample: n = 568 participants: Age ≥ 18 Male: n = 225 Female: n = 343 White participants: n = 331 Educated: n = 331 Years in neighbourhood: 7.2	Black: n = 138 Asian: n = 29 American Indian: n = 55 Hispanic: 93 Foreign born (n = 129)	Cross-sectional design Between-subjects comparison Dependent variables: (1) Park use frequency (2) Health benefits of parks (3) Barriers to park use Independent variables: Self-report measure of race/ethnicity	Likert-scale questionnaires (administered in four different languages (English, Spanish, Somali, and Hmong) Determine: (1) Park use frequency (2) Health benefits of parks (3) Barriers to park use	Foreign-born residents, Blacks and Hispanics highlighted the barriers to park use were: - Not feeling welcome - Cultural and language restrictions - Program schedule and pricing concerns - Facility maintenance and mismatch concerns	To improve the design of park strategies, services must address health disparities and remove the barriers that minorities and foreign-born communities face. Acknowledged limitations: The study only includes three neighborhoods in Minneapolis.
Derose et al., (2015) [83]	Los Angeles, California (USA)	Examine racial ethnic differences in park use and physical activity among adult residents	Momentary time sampling Sample: n = 7506 Participants: Age ≥ 18 White: n = 1594 Gender data not specified Residency: living within one mile of 50 parks in Los Angeles	Black: n = 807 Latino-English: n = 858 Latino-Spanish: n = 3735 Asian/PI/other: n = 512	Cross-sectional designs Between subjects’ comparison Dependent variables: -Park use—(defined as the number of times residents visited their neighbourhood park in 7 days) -Physical activity—(inactive—less than 50 min of physical exercise a week and active—more than 150 min) -Physical activity in parks -Social interactions in parks Independent variables: -Self-reported racial/ethnic group. -Park-level co-variates: -Park size (acres) -Park location in commercial vs. residential area (within one mile radius) -Proportion of households in poverty -Number of observed organised activities -Number of observed supervised activities -Association between neighbourhood racial-ethnic diversity and park use.	Park use—Measurement of park use questionnaire [90]. Physical activity—self-report measure created by researcher, based on government guidelines. Combined information of physical activity and park use created a “exercise vital sign” [91]. Use of parks for exercise—via self-report—(categorised by; (1) does not exercise, (2) exercises but not in parks, (3) exercises in park. Use of parks for social interaction—via self-report (categorised by; (1) does not go to the park (2) goes to park alone (3) goes to park alone but sees/meets others there or (4) goes to park accompanied. Association between neighbourhood racial-ethnic diversity and park use—measured via 2010 Census data and Simpson Index.	Regression models and bivariate analyses found that Blacks and English-speaking Latinos were less likely to report exercising outside of parks and more likely to socialise in parks. Blacks and Latinos were less likely to report exercising in all domains, compared to Whites. However, Spanish-speaking Latinos and Whites and reported using parks for exercise and socialising.	Urban parks appear to be an important resource for physical activity and socialisation, especially in Spanish-speaking Latino and Asian groups. More efforts are needed for other racial-ethnic minorities to experience the same benefits. Acknowledged limitations: data came from two cross-sectional surveys and causality could be inferred. Most measures were based on self-report and subject to recall and social desirability bias.
Dolash et al., (2015) [84]	San Antario, Texas (USA)	Assess factors associated with park use (in six parks) and physical activity among park users in predominantly Hispanic neighbourhoods	Opportunity sampling—to gain participants Sample: n = 2340) Unable to collect gender demographics Age: adults, adolescents (13 to 18 years old), and children (3–18)	Predominantly Hispanics	Mixed methods, Cross-sectional research design Two trained research assistants visited each study park across 3 days at the same time. -Assessed the park environment, presence of park features and park quality. Data collectors’ assessments were compared with measure interrater reliability. Additionally, observed physical activity at pre-determined play spaces. Play space activity was also scanned. Dependent variables: -Parks/play spaces available -Park use/physical activity -Motivations and barriers of park use Independent variables: -Ethnicity -Days of the week -Time (Afternoon/evening) -Play spaces (field, basketball, playground, tennis court, track/trail, fitness stations, baseball, and horseshoes) Park condition (nonrenovated: n = 4 and renovated: n = 2)	Park use—measured by direct observations (via the Parks and Play Spaces Direct Observation Tool; McKenzie et al., 2006) [92] Physical activity—measured by “computing energy expenditure by multiplying the total number of people in the play space, by a multiplicative constant, based on activity intensity. Physical activity energy expenditure scores represent the average kcal/kg/minute for each person in the play space, during the scan”. Parks/play spaces available—measured by an audit tool Park environment—measured by the Parks and Play Spaces Environmental Audit Tool [93]. Semi-structured interviews, for approximately 5 min (n = 51), assessing motivations and barriers of park use - Opportunity sampling—researchers approached people in the park who looked over 18 and relied on self-report to confirm this was true. - 33 interviews in non-renovated parks - 18 interviews in renovated parks	Renovated parks had higher Physical expenditure scores, than non-renovated parks. Basketball courts had a significantly higher number of vigorously active park users. Thematic analysis found four themes that explained lack of park use and physical activity: - Motivation to be physically active - Using the play spaces in the park - Parks as the main space for physical activity - Social support for using parks	Renovations to park amenities (increasing basketball courts, trail availability) could increase physical activity in low-socioeconomic-status populations. Acknowledged limitations: Cross-sectional design restricts understanding of causal mechanisms underlying the behaviour. The study data was collected in the winter and could not observe seasonal changes. Unable to collect data on race, ethnicity, or gender during observations.
Fernandez et al., (2015) [81]	Chicago, (USA)	Examine the difference in access to natural environments and acculturation levels among Latinos from two urban communities in Chicago.	Stratified sampling Sample: n = 376 Participants: Male: (n = 172) Women: (n = 204) Age: Adults Controlled for: -Education level -Income level -Generation status (e.g., born in U.S./Immigrant) -Average years spent in UK -Acculturation level	Mexican (n = 154) Puerto Rican (n = 20) Other Latin American country (n = 14)	Cross-sectional design Between groups comparison Questionnaires were randomly distributed to 392 Latino households, within two Chicago neighbourhoods. Measurements of; -Demographics -Participation in recreation activities -Acculturation -Access to natural environments	Self-report questionnaires for: - Demographics - Participation in recreation activities - Access to natural environments -Acculturation, measured by; The Bi-dimensional Acculturation Scale for Hispanics—BAS, [94]	Access to natural environments significantly increases the likelihood of recreation participation.	Increased access to natural environments for Latino communities is needed as a future intervention, to improve usage. Acknowledged limitation: Participants’ recall was used to measure recreation participation and therefore, participants could distort their true participation rate. Did not account for other factors related to accessibility (e.g., the ease of navigating and attractiveness of park features). Items related to acculturation (e.g., time spent in the US) not accounted for. Did not control for the ethnic origin of Latinos. Low response rates were also a concern.
Lee at al., (2015) [79]	Houston and Austin, Texas (USA)	To create and test an index to indicate availability and quality of physical activity (PA) resources (PARs) to examine associations between access to quality PARs and changes in PA. This assessment was completed on “minority women over time”. , Additionally, to determine whether this association differed in women from lower and higher income neighbourhoods.	Volunteer sampling Sample of Women (n = 410) Low/median/high income areas (Demographic information about ethnicity and household income, was adapted from the Maternal and Infant Health Assessment Survey [95].	African American Hispanic/Latina women	Randomised control trial Between groups comparison Longitudinal, 6-month intervention Participants attended a baseline time health assessment. Completed: -interviewer-administered questionnaire -Physical assessment -Packet (with more detailed questions) to complete before the next meeting one week later. This was also a “run-in” procedure to discourage less interested participants before randomisation. Women who completed the packet were randomized into one of two intervention group Experimental group: HIP procedures Two groups: (1) Physical activity group (2) Vegetable and fruit group Baseline: -Health assessment completed -Interviewer administered questionnaires -Physical assessment (given a packet to complete before the next meeting, approximately 1 week later). The 6-month face-to-face intervention included behavioural methods to promote group cohesion and to account for environmental factors contributing to health disparities. Women participated in team-building activities, environmental mapping exercises, and supervised walks or taste tests. After 6 months, women returned to complete identical health assessments. A subset of women (n = 59), completed a questionnaire and accelerometer to measure moderate to vigorous physical activity. Environmental cross-sectional and longitudinal individual level data determined the relationship between PAR and physical activity.	Self-report: The International Physical Activity Questionnaire (IPAQ) Measured self-reported physical activity, including work-related, transportation, domestic, and leisure-time physical activity (walking—moderate- vigorous-intensity physical activity, over the last 7 days). Physical activity resources were audited using the PARA (physical activity resource assessment [79]. Quality of physical activity resources were determined by a composite index (QPAR) of features, amenities, and incivilities. Body Mass Index (BMI)—calculated height and weight Physical activity was reported as metabolic equivalent of task (MET)- as minutes per week	Repeated measures ANOVA was used to show that; Women in neighbourhoods with lower quality of physical activity resources, showed small increases in physical activity, compared to women in neighbourhoods with higher quality park resources. These women also showed increased vigorous physical activity.	Access to better quality physical activity resources can help improve physical activity, regardless of neighbourhood income. However, physical activity resource quality is a distinctly important predictor of physical activity in ethnic minority women. Acknowledged limitations: The use of a median split to define income groups.
Mc Eachen et al., (2018) [80]	Bradford, (United Kingdom)	Explore associations between availability of, satisfaction with, and use of green space mental wellbeing among children aged 4 years in a multi-ethnic sample	Part of a wider study (Wright et al., 2012). Judgment sampling used. Sample n = 2594 Participants: Male: n = 1302 Female: n = 1292 Age: Children (4–5 years) Mothers of adult age.	White British: n = 740 South Asian: n = 1519 Other ethnicity (Black-African, Bangladeshi, and mixed race): n = 333	Cohort study/Longitudinal study (2007–2011) of 12,453 mothers recruited during their pregnancy and 13,776. Between subjects’ comparison	Access to residential green space (measured via the NDVI (normalised difference vegetation index; [96]. Self-report, Likert-scale questionnaires (completed by parents of the children) Assessed: (1) Satisfaction with green space (2) Use of green space	Unadjusted regression models were computed. Covariates were entered sequentially in logical blocks. Significant associations between availability of green space and behavioural difficulties in South Asian children living in deprived areas in the UK. More green space was associated with fewer behavioural difficulties. Ratings of satisfaction with green space was independently predictive of South Asian children’s mental wellbeing.	Poor quality parks and green spaces can discourage use by racialised communities. Satisfaction with green space is a more important predictor of wellbeing the quantity of green space. Public health professionals and urban planners need to focus on both the quality and quantity of green space to promote health, particularly in ethnic minorities. Acknowledged limitations: Mostly could not control for the effect of maternal mental wellbeing on children’s outcomes. Green space exposure was measured differently, throughout. May have caused mixed research Findings. Parental self-reporting could be subject to response bias.
Razani et al., (2020) [68]	Oakland, California, (USA)	Prior to park prescription, do park visitors face fewer sociodemographic barriers to park use, have more information about park location, increase park visits, increase park knowledge, and have more nature affinity as well as perceive less barriers to visiting parks.	Part of a wider study (Razani et al., 2016) Volunteer sampling and paid for participation ($40). Sample (n = 78) Caregivers and their children partook in the study. Participants: Female Caregiver: n = 68 Age—Children 4–17 years. Demographics/characteristic included: Age Race/ethnicity Immigration status Poverty level	African American (n = 52) Non-Latino White (n = 8) Latino (n = 12) Other (n = 5)	Randomised control trial Between-subjects comparison Families were randomized into two groups: -A supported group was invited to three organized group outings to parks. They received weekly text messages to remind them of the benefits of nature and encourage them to visit their parks. -Other group was free to visit parks on their own. Additionally, measured: -Baseline Park use per week (via self-report) -Park use over time. Caregivers reported on their weekly park visits at one and three months after receiving a park prescription. Predictors were: Knowledge, attitudes, and perceived access over time. These caregiver characteristics were measured at baseline, and at one and three months after receiving a park prescription	Baseline Park use per week—measured by self-report—caregivers reported their own park visit behaviour as well as the park visit behaviour of their children. Park use over time. Caregivers reported on their weekly park visits at one and three months after receiving a park prescription.	At baseline, White families were more likely to use parks, have prior knowledge park locations, value time in parks with family, and feel safe in their neighbourhood for their child to play. However, they were not more likely to value nature. After participants received a park prescription, park use increased as participants reported increased level of information about the location of parks, nature affinity and perceptions about time and resource availability. Non-white respondents and those who lacked neighbourhood safety were less likely to visit parks even once a week.	This study is the first to suggest that behavioural health theory will benefit the park prescription movement. This study suggests that the same populations at risk for health inequities in chronic illness are those who may be visiting parks less at baseline. Acknowledged limitations: Small sample size. Did not find significant differences in income level and frequency of park visits, Lack of precision in defining what park/nature is Study conducted in an urban center, so patients might not have access to parks with natural elements. No control over which parks participants visited.
Roe et al., (2016) [65]	United Kingdom (Midlands, Greater Manchester, and London)	How does general health differ between ethnic groups and what are the distinguishing health profiles? How does use and perceptions of neighbourhood environment/green space differ between ethnic groups? What demographics, social and physical attributes of place predict general health amongst different ethnicity general health profiles?	Quota sampling Sample: n = 523 Participants: Male: n = 40% Female: n = 60% Adults (over 16 years old) Age: 16+	White British: n = 114 Indian: n = 57 African-Caribbean: n = 63 Bangladesh: n = 89 Pakistani: n = 115 Other “BME”: n = 85	Cross sectional study. Household questionnaire explored the relationship between general health and individual, social, physical, and environmental predictors in deprived White British and British BME groups. Measured: Demographics General/physical health Social environment Neighbourhood environment Local green space	Demographics, general/physical health, social environment (perceptions of loneliness/place belonging) and neighbourhood environment, were measured via self-report Likert scales, based on local government recommendations (e.g., British Heart Foundation National Centre) Green space was measured via self-report, Likert-scale questionnaires assessing three items: (1) Safety (2) Attractiveness (3) Satisfaction	White British people found social characteristics of place (place belonging, levels of neighbourhood trust, loneliness) predictors of general health. Access to, use of and quality of urban green space was a significant predictor of general health in BME populations.	There needs to be better support for health in ethnic minorities, with the enhancement of green space in their environments. Acknowledged limitations: Included small numbers of racialised groups (e.g., Chinese, white European), had to aggregate some data, therefore losing distinctiveness across some ethnic groups, and generalisations were made. Limited in “best” health group (participants from Indian origin (n = 57)). Needed a larger sample size. Generalisations of findings may be difficult, due to differences in scale, context, culture, and geography. Subjective, self-report data is limited to bias. Used a standardized measures for general health to compare findings with the wider population. However, did not understand what health and wellbeing meant to different populations.
Schultz et al. (2017) [85]	Columbia, Missouri, (USA)	Evaluate the impact of street-crossing infrastructure modifications on park use/park-based activity in low income and African American communities.	Opportunity sampling Sample: n = 2080 Male: n = 1129 Female: n = 951 (preintervention) n = 2275 (post intervention) n = 2276 (follow up) Age: child 1–12 years, teen 13–20 years, adult 21–59 years, or senior 60+ years. Child (n = 574 pre; n = 555 post; n = 684 follow up) Teen (n = 362 pre; n = 441 post; n = 292 follow up) Adult (n = 1093 pre; 1159 post; 1177 follow up) Senior (n = 51 pre; n = 120 post; 121 follow-up)	African American Pre-intervention: n = 1483 Post-intervention: n = 1615 Follow-up: n = 1588 Other populations observed: White Other ethnicities were coded, but not included in the final data sample, e.g., Hispanic, Asian/unsure)	Natural/Observation experiment Within-subjects comparison Independent variable = Race (African American/White) Dependent Variable = Park use/physical activity after an installation of 26 signalised crossing	Observed Park use—measured by the modified System for Observing Play and Recreation in Communities (SOPARC; [92])—Uses momentary time sampling technique to collect systematic scans of park users to access park use within pre-determined activity areas. The 26 park activity areas were visually scanned left to right by trained observers and the codes representing park users and physical activity levels were recorded on a standardized form. The codes for physical activity also provided estimates of energy expenditure (EE) by assigning Metabolic Equivalents (METs) to recorded categories of physical activity following previous research (Sedentary = 1.5 METs, Moderate = 3 METs, Vigorous = 6 METs [97].	The installation of crosswalk signals improved park use overall.	Lack of safe access to parks may have been a barrier to park us in African American adult women. Acknowledged limitations: Infrastructure changes in the park (i.e., renovated fitness equipment and new walking trails) during the autumn 2013, led to site being unusable longitudinally. Intrapersonal and interpersonal factors are not addressed in this study. Extraneous variables (e.g., changes in crime rates), could not be controlled for. Observations of time frames based on natural rhythms of The community were not done. For example, a fourth (6:30 pm) time frame was Dropped, due safety concerns. Weather was measured only via temperature and no other measures to corroborate this (e.g., humidity). Observer bias in researchers.
Xiao et al., (2016) [86]	New York City (USA)	Examine the role of transportation in visiting national parks by three racial/ethnic groups.	Recruited via judgment sampling from an online survey Sample n = 600 Controlled for: -Age (all ages included) -Household income -Education Gender: Male: n = 128 Female: n = 172 White: n = 100	Hispanic n = 100 Black: n = 100	Cross-sectional study Between-subjects compassion Measured: -Sociodemographic information -Visitation to a National Park Service (NPS) Area in Last Two Years by Race/Ethnicity. - Barriers to Visiting National Parks - Agreement with Transportation Incentives	General Population survey Measured (all via self-report measures): - Socioeconomic information - Issues of visitation status - Barriers to visiting national parks - Likelihood of respondents increasing their visitation if they had transportation incentives (e.g., Faster transport, less expensive transportation, more/better forms of public transport (e.g., shuttle buses in parks), More opportunities to walk/bike within and to/from parks, more information (maps, etc.) about transport to/from parks, more parking	37% of White residents visited national parks, within the last two years, compared to 31% Hispanics and 23% Blacks. Barriers to access to national parks were divided into three categories (comfort and safety, expense, and accessibility). Hispanics reported significantly higher comfort and safety barriers, compared to Whites. Black Park visitors reported comfort and safety a higher barrier than Black non-visitors. Blacks and Hispanics reported expense (cost of transport, food, entrance fees and lodging) as their greatest limitation to park access. Ethnic minorities also reported accessibility (means of transportation) as a barrier, compared to White counterparts. Hispanics were also more accepting of incentives (e.g., less expensive, faster, and better means of travelling to the parks), than Whites.	The barriers highlighted link with the marginality hypothesis and the role of transportation incentives. Transportation incentives may be more crucial for attracting a more representative audience, to national parks. Acknowledged limitations: Online survey led to lower response rates. Well-educated respondents were overrepresented. Future research could “explore the differences in outdoor recreation preferences among racial/ethnic groups and the relationship between barriers to visitation and recreation preferences”.

### 3.2. Critical Appraisal

The three studies appraised via the Kmet et al., (2004) checklist [76] obtained quality appraisal scores that ranged from 64.3% to 95.5% (M = 77.1%). The remaining studies, appraised via the Downes et al., (2016) checklist [75] scored between 77.5% and 85.0% (M = 81.5%).

All studies showed strengths in reporting clear research aims, appropriate methods, clearly defined variables, well defined target populations, with samples taken from appropriate sources. All successfully described/justified analytic methods and reported results with appropriate statistical analyses (where appropriate), and concluded the results appropriately whilst acknowledging limitations.

Regarding the distinct criteria, not all studies appraised by the Kmet et al., (2004) [76] checklist required randomisation, as one of the studies used a cohort design [80]. Only one study was awarded full marks, for reporting random allocation of participants in a randomised control trial, and no study reported blinding their investigators/participants. However, two studies used robust/standardised outcome measures. 

For those studies appraised by the Downes et al., (2016) [75] checklist, none of the papers reported power calculations to justify their sample size. However, the lack of an effect size is possibly explainable for the longitudinal study [83] given the number of repeated assessments and level of missing data [98].

Internal consistency was also unclear in 71.4% of the reviewed studies. Only one study [80] reported Cronbach’s Alpha to establish internal consistency within the Strengths and Difficulties Questionnaire. However, these were the only measures tested for internal consistency, in both studies.

None of the studies scrutinised by Downes et al.’s., (2016) [75] checklist reported examination of non-responders. No study noted whether ethical guidelines were followed, and none clearly reported whether they had gained consent from their participants, in contrast to those three studies assessed via the Kmet et al., (2004) [76] checklist.

### 3.3. Overview of Findings/Barriers Highlighted (Figure 2)

#### 3.3.1. Interpersonal

##### Feeling Unwelcome/Out of Place

Two of ten studies revealed that appraisals of not ‘feeling welcome’ was a barrier to accessing green space for racialised participants. Das et al. (2016) [82] reported this for foreign-born residents (16%), Black ethnic/minority groups (15%), and Hispanics (13%), as did Roe et al., (2016) for people from African-Caribbean, Bangladeshi, Pakistani and “other B[A]ME backgrounds”, who also reported lower feelings of place belonging within their neighbourhoods. This factor also appeared a significant predictor for “poor health” in these groups.

**Figure 2 ijerph-20-00745-f002:**
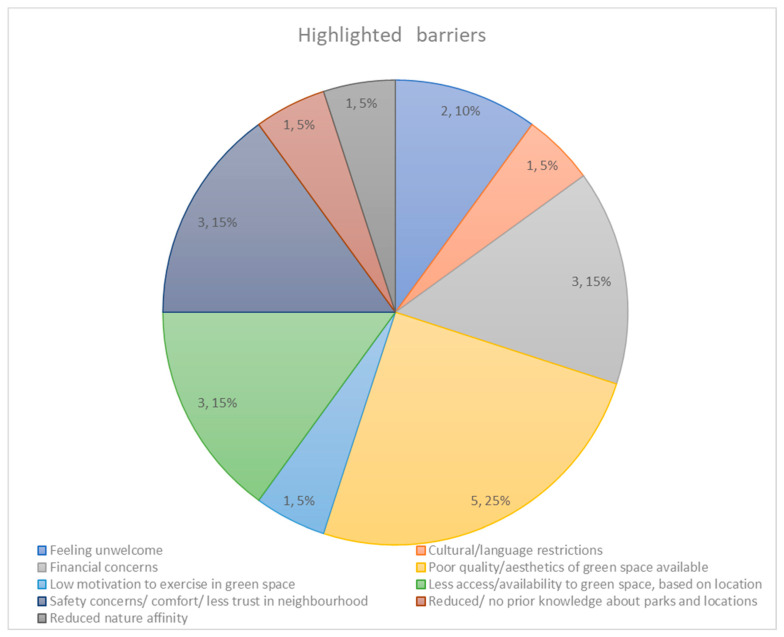
Provides a visual overview of the themes of the main barriers experienced by racialised communities in relation to green space access.

##### Cultural/Language Restrictions

Das et al., (2016) [82] found that foreign-born respondents were more likely to perceive ‘language barriers’ (OR = 3.82, *p* < 0.01) and ‘cultural restrictions’ (OR = 4.45, *p* < 0.01), as a barrier for park use suggesting for individuals not born in the country of residence, reduced access to green spaces, may be due to perceived language barriers.

##### Low Motivation to Exercise in Green Space

Low motivation appeared to be another barrier for park use. Dolash et al., (2015) [84] found that Hispanic, park users were most motivated to go to the park and be physically active if it involved their children (either taking them there to play or being healthier for their children).

#### 3.3.2. Practical 

##### Financial Concerns

Three studies reported financial concerns as barriers to USA park use for racialised communities. Through regression analyses, Das et al., (2016) [82] found that Black minority groups reported ‘program pricing concerns’ 1.89 times more (OR = 1.89, *p* < 0.05) than all other categories. In Xiao et al., (2016)’s [86] study, ANOVA and binary logistic regression models revealed that ‘Blacks and Hispanics’ reported expense (accounting for 20% variance, α = 0.91), as their biggest barrier to visiting national parks. The highest loading factors were high transport, food, entrance fees and lodging costs. Razani et al., (2020) [68] reported that being unable to afford to travel to parks was the most common barrier for park use in non-White participants (53%).

##### Poor Quality/Aesthetics of Green Space

Poor quality/aesthetics of green space was a barrier noted in half of the included studies. Das et al., (2016) [82], found that Hispanics (n = 20, M = 21.74) were more likely to perceive ‘lack of maintenance’ (OR = 2.20, *p* < 0.05) and ‘facilities not meeting needs’ (OR = 1.94, *p* < 0.1) as barriers for park use compared to non-Hispanics. Observational data by Dolash et al., (2015) also found that Hispanics showed higher energy expenditure scores in renovated parks (M = 0.086 ± 0.027) than non-renovated parks (M = 0.077 ± 0.028; t = −3.804; *p* < 0.01).

When comparing physical activity (PA) with the quality of physical activity resources (QPAR), in African American and Hispanic/Latina women, Lee et al., (2015) [79] found that women in neighbourhoods with lower QPAR scores showed smaller increases of PA (M Δ = 327.8 metabolic equivalent of task [MET]-min/wk), compared to women with higher QPAR scores, with larger increases in self-reported PA (M Δ = 709.8 MET-min/wk). Additionally, there was a significant interaction between changes in leisure-time PA, QPAR score, and number of PARs (*p* = 0.049).

Among South Asian children, Mc Eachen et al., (2018) [80] found that satisfaction with green space was significantly associated with fewer behavioural difficulties (β −0.59 [95% CI −1.11 to −0.07]) and more prosocial behaviour (0.20 [0.02 to 0.38]); interestingly, similar associations were not observed among White British children.

Self-report data also highlighted that perceived neighbourhood green space quality (i.e., safety, attractiveness, satisfaction with urban green space) is a consistent predictor for general health across the ‘worst’ health group, i.e., African-Caribbean, Bangladeshi, Pakistani origin and other ‘BME’ groups, [65].

#### 3.3.3. Environmental

##### Less Access/Availability to Green Space, Based on Location

Self-report evidence revealed disparities in green space access for racialised individuals. Fernandez et al., (2015) [81] found that areas in Chicago with high Hispanic/Latino populations (e.g., Little Village) had less access to natural environments (e.g., a back yard). Equally, there was a significant difference in reported access to backyards and significantly less green space access (χ^2^ (1, N = 376) = 14.18; *p* < 0.001).

Xiao et al., (2016) [86] also found that one of the highest loading factors of accessibility barriers for racialised groups were difficulties getting to National Park Service areas, as it “takes too long”. Additionally, among the three racial/ethnic groups Hispanic respondents reported the highest levels of agreement with this barrier compared to Whites (η = 0.14).

##### Safety Concerns/Comfort/Less Trust in Neighbourhood

Evidence in three studies [68,85,86] appears to support the belief that racialised individuals limit park use due to concerns about safety levels, comfort, and general distrust of park location. Razani et al., (2020) [68] compared individuals who visited a park in the week prior to the study taking place to those who did not. They found that park visitors (who visited a park at least once) were more likely to report living in a safer neighbourhood for their children to play in (Figure 3).

Xaio et al., (2017)’s [86] study evidenced that the differences between park visits for Hispanics and Whites were significant for perceived comfort and safety (η = 0.11). Additionally, potential improvements in comfort and safety had a positive effect on Black respondents’ preference for visitation (OR = 3.24). Evidence also highlighted an increase in park use after installing signalized crosswalks nearby (including for Black participants), suggesting the importance of safe access, to and from parks [85].

#### 3.3.4. Experience/Knowledge

##### Prior Knowledge about Parks and Locations

Increased knowledge about park location was predictive of increased park use, in Razani et al., (2020)’s [68] study. They found that as parents’ park knowledge increased, their visits to parks increased by one park visit over three weeks [(95% CI 0.05, 0.50), *p* = 0.016]. The authors acknowledged this was a small increase but predicted that this could promote a clinically significant increase over several months.

##### Nature Affinity

Razani et al., (2020) [68] reported low nature affinity as a barrier, whilst accounting for seasonal effects (by assessing during both the summer and winter months) they found evidence for a positive association between nature affinity and improved park visits. As nature affinity increased (assessed via a seven-point scale) park visits increased by one visit every three weeks. 

## 4. Discussion

This review aimed to explore the barriers to green space access for racialised individuals and families, and identified diverse psychosocial and economic barriers (interpersonal, practical, and environmental) potentially explaining why racialised individuals and families access green spaces less than White individuals. This narrative review is one of the first to summarise both USA and UK studies reporting on quantitative data capturing psychosocial barriers to green space access for racialised individuals and families, and has utilized the USA-originating EJF framework. This framework has been predominantly applied in the USA to understanding BIPOC disparities in green space access. This review has applied the EJF framework to understand barriers to access in both the UK and USA for non-White populations and the environmental injustices that racialised individuals and families experience. Despite notable cultural differences between the UK and USA, the results suggest that the EJF framework is a useful theory to understand the commonalities between experiences of both racialised and BIPOC populations in both countries regarding barriers to green space access.

Our findings may thus offer policy makers and commissioners preliminary guidance for prioritising means to improve green social prescribing referrals for racialised individuals and families. Synthesis from this review may also inform green social prescribing via the link worker programme within primary care [17], offering information on how to consider race and the potential equality impacts when designing and commissioning programmes.

### 4.1. Interpersonal

Cultural barriers (e.g., language and feeling unwelcome), to green space access appeared prominent and echoes previous evidence highlighting the racial disparities in green space access due to inequitable care access [41] and a lack of cultural adaptation within communities [42]. All reviewed studies reported on samples from countries with majority White populations where structural racism (e.g., institutional racism), [99,100], health inequalities (as seen with the treatment of racialised groups [e.g., African-Caribbean, Bangladeshi, Pakistani] during the COVID-19 pandemic), [10] and overt discrimination (e.g., bigoted name calling), [101] is often experienced by racialised groups [102]. Our review findings may suggest there is systemic rural racism [103] that further alienates racialised individuals and families from green spaces and contributes to feeling “out of place” in these settings. Policy makers could initiate diversity-friendly schemes to improve engagement in green spaces for racialised communities, for example through introducing multiple languages on signs and additional prayer spaces in parks [104] to promote inclusivity.

### 4.2. Practical

Financial concerns too were highlighted as a major barrier for accessing green spaces for racialised individuals and families, corroborated by self-report and government data. That socio-economically deprived individuals access fewer green spaces [31] due to an inability to afford travelling to/entering these spaces (e.g., transportation, entrance fees and lodging costs) has been previously advanced. This can be understood as a manifestation of a hierarchy of needs through which those who are structurally poorer prioritise finances for more “urgent concerns” such as food/shelter [105], and are thus unable to justify spending key finances on accessing green spaces. Such evidence argues for targeted funding in services to support racialised individuals/families access green, socially prescribed activities to improve equitable access for all.

### 4.3. Environmental

Factors predicting green space accessibility for racialised individuals and families included safety, comfort, park quality/aesthetic, and park proximity. Results from this review showed that South Asian children in England were less likely to show behavioural difficulties if satisfied with their green spaces (parks) [80]. Park visitors were more likely to report living in a safer neighbourhood for their children to play in [68], and this is unsurprising given racialised families are likely to live in the most deprived neighbourhoods, furthest from green spaces [43,106]. Data from the Police UK Indices of Multiple Deprivation (2022) [107] reported that 80% more crimes were recorded in the most income-deprived neighbourhoods. Arguably, racialised individuals and families may avoid green spaces because of the heightened threat of crime and given heightened salience given recent violent attacks on racialised women in green spaces within in the UK [108].

Green spaces should afford a safe space for individuals [109] and research has shown the impact of quality green spaces on reducing crime rates [110]. However, this review highlights that environmentally disadvantaged, racialised individuals and families residing in unsafe, poorer quality neighbourhoods are less comfortable engaging with green spaces, resonating with findings that BIPOC youth accessed green spaces less due to fear [36]. Arguably, initiatives to mitigate fear are likely to involve greater policing of parks in concert with urban planning to ensure good quality, safe green spaces in socially deprived areas, to improve comfort levels and access for all.

### 4.4. Green Gentrification

Our review findings further underline the impact of gentrification on marginalised/racialised BIPOC within the USA [68,79,81,82,83,84,85,86], and suggests its relevance in a UK context creating further environmental injustice [64,68]. Safety is revealed as a key concern in this review, alongside access to good quality green space. Social graces theory highlights how multiple intersectionalities can increase vulnerability to adversity [111]; individuals from racialised backgrounds may live in socioeconomically deprived communities, as well as derive from a socio-economically lower classes and be unable to afford the new and safer green spaces created through gentrification. Urban planning needs to urgently consider its role in reducing disadvantage to of racialised individuals and families and reducing further environmental injustice.

Whilst the overall results are helpful in highlighting clear barriers, it is important to recognise that many studies included were limited by design, were mainly cross sectional and lacked controls or longitudinal data, making it difficult to infer causality or understand how barriers to access may change over time [112]. Additionally, most studies used self-report measures, which despite being important for understanding personal experiences also risk influence of social desirability/interpreter bias and demand characteristics. Lastly, many studies failed to report gaining consent for participation, which may raise ethical concerns. It is thus difficult to ascertain the reliability and internal validity of most studies and results should be viewed with caution. Nonetheless, this review is the first of its kind to synthesise peer-reviewed, quantitative research reporting the psycho-socioeconomic barriers to green space access expressed by racialised individuals and families/BIPOC from their own experiences and perspectives in the both the UK and USA. Whilst fewer in number, UK publications point to similar difficulties regarding environmental injustice for racialised individuals and families that is further reinforced by additional intersectionalities (e.g., socioeconomic status and class). The lack of equitable green space access for all combined with practical, interpersonal and environmental barriers for racialised individuals/families and BIPOC means that access to green social prescribing may be more difficult for these individuals. Directed interventions to address these factors could help BIPOC and racialised individuals and families access green spaces and green social prescribing and the EJF could fruitfully inform future studies exploring environmental injustices, in the UK, the USA and further afield.

### 4.5. Strengths and Limitations

This review used multiple databases to yield searches, which increased breadth of relevant articles. Additionally, our focus on a wide range of racialised groups (African, Caribbean, “Black”, Hispanic/Latino, west/south/south-east Asian), encompasses many racialised individuals/families in the UK and USA offering ecological validity and generalisability. The quality appraisal tools used to evaluate the studies are routinely utilized, systematic and robust, improving confidence in the external validity of the review with scoring for both appraisal tools supported by a standardised manual. Nevertheless, quality appraisal was undertaken by the lead author conferring a risk of subjective bias which may affect the overall internal reliability of the appraisal.

While international studies were included in this review, elicited papers were published in English and limited to two countries (USA and UK) with majority White populations. Since different terms may encompass ethnicity and race, study identification and comparisons were challenging. For example, African American may differ from Black British experiences and a lack of precision regarding the reporting of race and ethnicity, along with burgeoning terminologies [113] this may constrain generalisability of findings.

Despite the large sample sizes, researchers only focused on specific groups of racialised individuals/families (African, Caribbean, “Black”, Hispanic/Latino, South Asian and White), and other than Razani et al., (2020) [68] offered no power calculations. The included studies focused on only one type of green space (parks/recreational spaces), which arguably reduces the generalisability of the findings to other NBIs on a global scale.

Our review examined quantitative research, with certain studies using a mixed methods approach to extrapolate data. This review only reported the quantitative findings. While experimental data was useful in gaining an understanding from a large representative sample, inductive approaches may be useful in understanding intrinsic motivators and barriers for racialised groups, when accessing green space. Future empirical research using longitudinal data is arguably needed to explore how psychosocial and economic barriers to green space access for racialised individuals and families may vary over time. Respectively, reviews could examine qualitative studies which may offer richer, nuanced data on the contextual mechanisms of engagement.

## 5. Conclusions

To conclude, this review is the first to consider the barriers for green space access for racialised individuals/families. Findings can help inform the design and commissioning of green social prescribing initiatives in the future to improve inclusivity for racialised individuals/families, and can direct future research to further examine inter-racial differences between racialised individuals/families, through more robust controlled study designs and participant-led qualitative studies.

## Figures and Tables

**Figure 1 ijerph-20-00745-f001:**
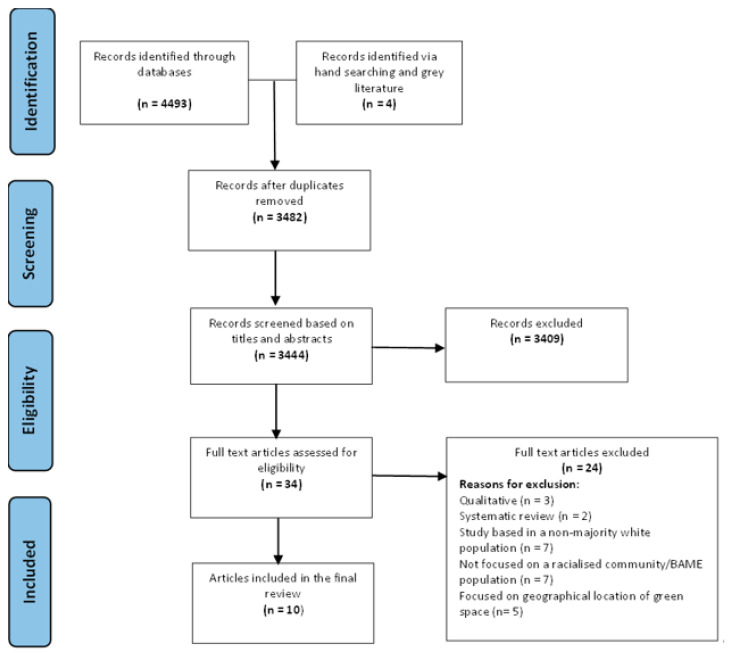
PRISMA Diagram.

**Figure 3 ijerph-20-00745-f003:**
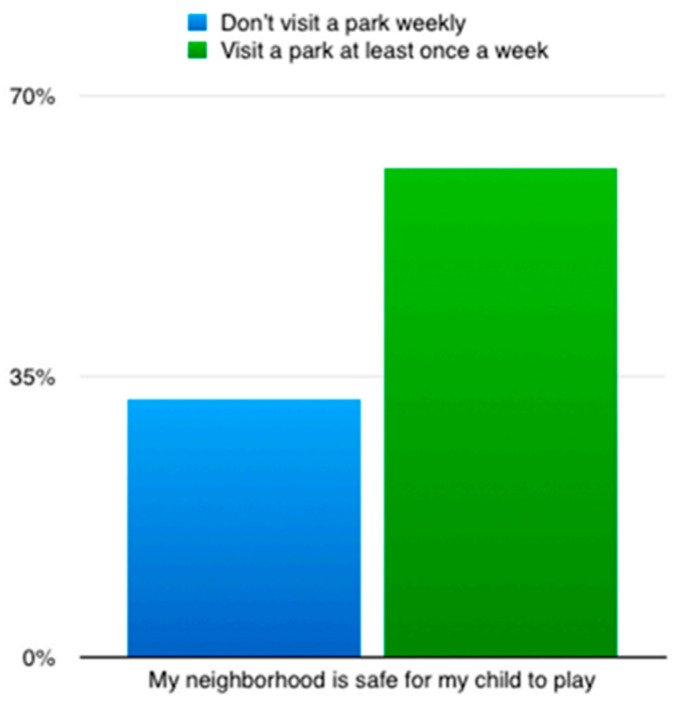
Bar graph showing perceived neighbourhood safety in Razani et al., (2020)’s [68] study.

**Table 1 ijerph-20-00745-t001:** Operationalisation of search terms using PICO framework.

PICO Criteria	Operationalisation of Search Terms
**Population**	Individuals who originate from a racialised community
**Intervention**	Green space/outdoor recreation access
**Comparison**	Within-subject comparison (pre/post intervention) or between-subject comparison with white counterparts
**Outcome**	Improved green space access and/or physical/mental health benefits

**Table 2 ijerph-20-00745-t002:** Inclusion criteria and exclusion criteria.

Inclusion Criteria	Exclusion Criteria
Participants: Racialised individuals and families of any age. - Definitions of racialised communities varied, depending on the country. - Racialised communities were identified from majority white, high income [69], developed countries (defined by a country not requiring developmental aid, under the rules of a multilateral or bilateral agency; [70]. - A broad range of terms for racialised individuals/families (see Table A1, Table A2, Table A3 and Table A4) were used for the search, to identify all available literature in this area. Majority sample from racialised communities (n > 50). All ages, genders, and socio-economic status’	Literature that did not focus on racialised communities.
Design: Experimental, quantitative research design for comparison of results through time and between conditions. Mixed methods studies were eligible but only quantitative data was synthesised.	Literature written in non-English language. Non-published papers, dissertation theses and conference papers. Systematic reviews and meta-analyses. However, relevant studies extracted from the reference list from these reviews were included. Solely qualitative research (excluded for clearer synthesis)
Intervention: Any active involvement with green space [71]. - Recreation use to improve mental/physical health (e.g., exercise, walking, playing), aesthetic appreciation/inspiration for culture, art/design, tourism, spiritual experience, sense of place. - Involvement could take place in diverse types of green spaces (e.g., community woodlands, urban parks, gardens, wetlands, [72]. Activity in green space of any frequency/duration. Individual or community level access.	Research that focused on urban planning/urban forestry, with a geographical focus
Outcome: Measures that evaluated: barriers, frequency, quality, and views of green space for racialised communities.	Research without complete data

**Table 3 ijerph-20-00745-t003:** Data extraction tool [74].

Data Extraction Category	Column Heading
Publication details	First author, year, type of publication, brief study aims, title of article
Study design	Study location, study design, comparison, randomisation, blinding, effectiveness of blinding.
Participant characteristics	Study sample description, total sample size analysed, power calculation for sample size, participation rate, definition of low income, sample age group, age, gender, total N female, total % female, ethnicity, education, employment, living situation, health status, annual income, sampling/how recruited, descriptive data of demographics, notes.
Racialised communities	Ethnic/cultural background of participants included in the study.
Intervention	Experimental intervention, intervention detail, intervention type, aim of intervention, who provided intervention, setting, key characteristics of setting, intervention frequency, intervention duration, control intervention (if used), any theoretical frameworks used to develop intervention, notes.
Outcomes	Method of data collection, who collected data, within/between subjects differences, primary outcome(s), primary outcome measure(s), outcome measure(s) detail, outcome measure respondent, validity/reliability of measure, timepoints, statistical tests used, comparison reported, effect size, description of findings, secondary outcome(s), secondary outcome measure(s), outcome measure(s) detail, outcome measure respondent, validity/reliability of measure, timepoints, comparison reported, effect size, description of findings, intervention compliance, intervention other findings.
Key observations	Discussion point, strengths, limitations, user/stakeholder involvement in design/conduct of study, theory/conceptual models used, critical appraisal points, other comments.

## Data Availability

Data is contained within the article.

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
