# Peer review of "Examining Psychosocial and Economic Barriers to Green Space Access for Racialised Individuals and Families: A Narrative Literature Review of the Evidence to Date"

_ijerph, 2022, doi:10.3390/ijerph20010745_

Round 1
Reviewer 1 Report
The topic is exciting, and the proposed objective of the study has significant practical implications.
However, the announced aim of the research, i.e. "to identify the psycho-socioeconomic barriers to green space access for radicalized individuals/families", within the framework of a systematic review, seems daunting. Illuminating what surfaced in the analyzed articles might be one of the possible options by that moment in that respective field.
The systematic review addressed peer-review papers from JAN-FEB 2022, although all ten articles analyzed were published between 2015-2020. Would you please clarify this detail?
It might be difficult for those unfamiliar with the NHS programs and ecosystem to grasp the true meaning of social prescribing used by authors. Maybe a short comment on this could make all the difference.
It could be very instructive for the journal's readership to know how much grey literature and hand-searching was performed. A short explanation only to have the whole picture of the methodological part and efforts done by the authors to secure a complete set of initial data for the analysis.
The paper is written from a medical perspective, taking into account also some of the theoretical perspectives used in analyzing the racial disparities in green space access (EJF, for instance). One would expect to find those frameworks used in discussing the findings, not to mention examples of behavioural interventions and public policy successes in this field.
The research design was followed scrupulously for this research paper.
On the issues mentioned above, minor clarifications are needed, in my opinion, to make the paper more attuned to the readership of the IJERPH.
Author Response
FEEDBACK: However, the announced aim of the research, i.e. "to identify the psycho-socioeconomic barriers to green space access for radicalized individuals/families", within the framework of a systematic review, seems daunting. Illuminating what surfaced in the analyzed articles might be one of the possible options by that moment in that respective field.
RESPONSE: Thank you so much for your positive and useful feedback. The review aims are highlighted from line 199 – 204 and has now been amended slightly to “Identify the psychosocial and economic barriers to green space access, for racialised individuals and families/ BIPOC within UK and US contexts.” and “ Results from this review can help to inform green social prescribing interventions.”
FEEDBACK: The systematic review addressed peer-review papers from JAN-FEB 2022, although all ten articles analyzed were published between 2015-2020. Would you please clarify this detail?
RESPONSE: line 224 – 225, has now been amended to reflect the accurate timings of the search.
FEEDBACK: It might be difficult for those unfamiliar with the NHS programs and ecosystem to grasp the true meaning of social prescribing used by authors. Maybe a short comment on this could make all the difference.
RESPONSE: Thank you so much for highlighting the need to explicitly define what social prescribing looks like within UK/NHS contexts. We have addressed this in lines 66-72, where the authors highlight what social prescribing is, in relation to NHS programmes.
FEEDBACK: It could be very instructive for the journal's readership to know how much grey literature and hand-searching was performed. A short explanation only to have the whole picture of the methodological part and efforts done by the authors to secure a complete set of initial data for the analysis.
RESPONSE: Thank you for highlighting the need for more detail with regards to the hand searching and grey literature and you requested a short explanation of the methodology and efforts done by authors to secure a complete set of initial data for the analysis. This is rectified in lines 227-230, with additional comments made about how inter-rater reliability was ensured.
FEDBACK: The paper is written from a medical perspective, taking into account also some of the theoretical perspectives used in analyzing the racial disparities in green space access (EJF, for instance). One would expect to find those frameworks used in discussing the findings, not to mention examples of behavioural interventions and public policy successes in this field.
RESPONSE: Thank you for illuminating the need to incorporate the theoretical underpinnings into the discussion of the paper. In lines 169 – 189 the authors have addressed the issues of gentrification that further reinforce the EJF in UK and US contexts. Additionally, in the discussion section, 482 – 491, the authors highlight the nuanced nature of this review using the EJF framework to understand barriers to green space access in the UK and US for non-White populations. Additionally, line 547 to 557 evidences discussion about the EJF in accordance with gentrification and how this review reinforces this.
FEEDBACK: On the issues mentioned above, minor clarifications are needed, in my opinion, to make the paper more attuned to the readership of the IJERPH.
RESPONSE: addressed as above.
Reviewer 2 Report
This manuscript presents documentary research which explored psychosocial and economic barriers to green space access for racialised individuals and families.
I am really concerned about the quantity of literature used for the data analysis. Only 10 literatures included in the analysis, and most of them conducted in the USA, and a few studies conducted in UK. The issues of racism are quite different in each country and region. Therefore, the results of this research can be hardly generalizable. Therefore, this research should focus only green spaces access of racialised individuals and families in UK or USA contexts.
I think that this research issue still needs to be conducted based on primary surveys (empirical studies) as the number of studies based on a primary survey are still scarce. Most important, this type of study needs empirical prove rather than documentary surveys which the number of studies are very scarce.
Since only few literatures were included in this study, I think the results still needs more literature supports, otherwise, empirical studies still need to be conducted to explore psychosocial and economic barriers to green space access for racialised individuals and families.
Some parts of the results of this study were generated from only the data from one literature. For instance, the effect of cultural/language restrictions on green space access of racialised individuals and families is only reported in the study of Das et al., (2016).
Based on documentary study, using only one or a few studies to generate a qualitative result makes the result unreliable and unconvincing.
Same as the effect of cultural/language restrictions on green space access of racialised individuals and families, the effect of low motivation to exercise in green space on green space access of racialised individuals and families is also only reported in the study of Dolash et al., (2015) [72].
The effect of financial concerns and less availability to green space on green space access of racialised individuals and families, is also reported by only 2 studies.
The effect of knowledge and nature affinity is also reported in only one study.
I feel like the result section in this study is more like a literature review, and the results of this study still need an empirical study to prove its claims.
Another comments:
Line 123-125, please explain in details why individuals residing in socioeconomically deprived communities have less access to good quality green space than those residing in higher socioeconomic communities.
Line 125, What do you mean by good quality green space?
Author Response
FEEDBACK: This manuscript presents documentary research which explored psychosocial and economic barriers to green space access for racialised individuals and families.
I am really concerned about the quantity of literature used for the data analysis. Only 10 literatures included in the analysis, and most of them conducted in the USA, and a few studies conducted in UK. The issues of racism are quite different in each country and region. Therefore, the results of this research can be hardly generalizable. Therefore, this research should focus only green spaces access of racialised individuals and families in UK or USA contexts.
RESPONSE: Thank you for your comments. We agree there is a limited number of literatures included in the analysis, which impacts the level of generalisability with regards to other countries. We highlight this limitation within the discussion section (line 590 – 595). Despite this, we have acknowledged the international context within the introduction (line 155-162).
Additionally, the focus of this study was specifically on western/White majority countries (hence UK and US focus), as authors wanted to highlight the barriers for racialised individuals within White majority contexts (e.g., UK, US and Australia). Due to the limited number of quantitative studies available No Australian studies met the inclusion criteria. Therefore, the review nuanced its focus to incorporate the EJF as the theoretical framework currently used in US studies and examined its applicability to UK studies.
The focus on the UK is because this review sought to understand how the findings could be used as evidence for the tailoring of green social prescribing programmes within the UK context to be more inclusive to racialised individuals and its part of a larger NIHR funded study. Additionally, the focus on the UK and US only has been justified more within the introduction, utilising US and UK specific examples of gentrification and the EJF (line 165 – 189).
Equally a lot of the research that focuses on the UK favoured geolocation rather than exploring psycho-socioeconomic barriers to green space access, therefore these alongside qualitative studies were excluded from the review. This aspect is evidenced in PRISMA diagram (line 247).
FEEDBACK: I think that this research issue still needs to be conducted based on primary surveys (empirical studies) as the number of studies based on a primary survey are still scarce. Most important, this type of study needs empirical prove rather than documentary surveys which the number of studies are very scarce.
Since only few literatures were included in this study, I think the results still needs more literature supports, otherwise, empirical studies still need to be conducted to explore psychosocial and economic barriers to green space access for racialised individuals and families.
RESPONSE: Thank you so much for your feedback and we acknowledge the limited quantitative research in this area. This could be an area of focus for future studies (highlighted in the discussion section), however, as this is a literature review an empirical study has not been done. However, we highlight the need for more empirical research to explore the issues raised (line 605 -607).We have included that it is a narrative review in the title to be explicitly clear “Examining psychosocial and economic barriers to green space access for racialised individuals and families: A narrative literature review of the evidence to date”.
FEEDBACK: Some parts of the results of this study were generated from only the data from one literature. For instance, the effect of cultural/language restrictions on green space access of racialised individuals and families is only reported in the study of Das et al., (2016).
Based on documentary study, using only one or a few studies to generate a qualitative result makes the result unreliable and unconvincing.
Same as the effect of cultural/language restrictions on green space access of racialised individuals and families, the effect of low motivation to exercise in green space on green space access of racialised individuals and families is also only reported in the study of Dolash et al., (2015) [72].
The effect of financial concerns and less availability to green space on green space access of racialised individuals and families, is also reported by only 2 studies.
The effect of knowledge and nature affinity is also reported in only one study.
I feel like the result section in this study is more like a literature review, and the results of this study still need an empirical study to prove its claims.
RESPONSE: Thank you for your feedback and we appreciate the limited number of studies to support each section. We chose to use a narrative synthesis approach and to separate each section to categorise the main barriers. We have added in the caveats about the evidence only pertaining to one study in certain lines 276 – 278.
Another comments:
FEEDBACK: Line 123-125, please explain in details why individuals residing in socioeconomically deprived communities have less access to good quality green space than those residing in higher socioeconomic communities.
RESPONSE: Thank you for expressing your need to clarify why individuals residing in socioeconomically deprived communities have less access to good quality green space than those residing in higher socioeconomic communities. We would like to highlight that this has been addressed in section 1.7 when it highlights gentrification. This explains why socioeconomically deprived populations are more likely to live in areas with less quality green space, as BIPOC/racialised individuals and families are also less likely to live in the most socioeconomically deprived areas (highlighted further in line 551, via the inclusion of social graces theory).
FEEDBACK: Line 125, What do you mean by good quality green space?
RESPONSE: Thank you for the need to clarify the meaning of quality green space within this review. You will find in line 93-94.
Reviewer 3 Report
Dear authors.
It is my pleasure to review your manuscript -systematic review- entitled: ” Examining psychosocial and economic barriers to green space access for racialized individuals and families”, ijerph-1984802, sent to the Journal of Environmental Research and Public Health.
The review stablished two aims: 1) to identify the psychosocial barriers to green space access, for racialized communities, and 2) to understand what cultural adaptions could be made, to help support racialized groups in accessing green social prescribing.
Although the methodology followed could be acceptable, the articles selected and their analysis cannot provide information of good quality to get, not only sound, but reasonable conclusions.
As you said in the conclusion section: “… it is important to recognise that many studies included were limited by design, were mainly cross sectional in nature and lacked controls or longitudinal data; making it difficult to infer causality or understand how barriers to access may change over time. Also, most studies used self-report measures, which despite being important for understanding personal experiences, also risk influence of social desirability/interpreter bias and demand characteristics. Lastly, many studies failed to report gaining consent for participation, which raises ethical concerns.
In a good number of the studies, there is no clear control group. There are not comparisons between groups, and when there are, statistical differences are not significant, or are not clearly mentioned. There is no gender information; the study designs are very different, disperse, being many cross-sectional. Different population in terms of age. It is difficult to ascertain the reliability and internal and external validity of the results. Conclusions are probably based on previous hypotheses or the results and conclusions of other studies, included in the review or included in the text, but are not frequently obtained through the results shown in the tables.
Author Response
FEEDBACK: It is my pleasure to review your manuscript -systematic review- entitled: ”Examining psychosocial and economic barriers to green space access for racialized individuals and families”, ijerph-1984802, sent to the Journal of Environmental Research and Public Health.
The review stablished two aims: 1) to identify the psychosocial barriers to green space access, for racialized communities, and 2) to understand what cultural adaptions could be made, to help support racialized groups in accessing green social prescribing.
Although the methodology followed could be acceptable, the articles selected and their analysis cannot provide information of good quality to get, not only sound, but reasonable conclusions.
As you said in the conclusion section: “… it is important to recognise that many studies included were limited by design, were mainly cross sectional in nature and lacked controls or longitudinal data; making it difficult to infer causality or understand how barriers to access may change over time. Also, most studies used self-report measures, which despite being important for understanding personal experiences, also risk influence of social desirability/interpreter bias and demand characteristics. Lastly, many studies failed to report gaining consent for participation, which raises ethical concerns.
In a good number of the studies, there is no clear control group. There are not comparisons between groups, and when there are, statistical differences are not significant, or are not clearly mentioned. There is no gender information; the study designs are very different, disperse, being many cross-sectional. Different population in terms of age. It is difficult to ascertain the reliability and internal and external validity of the results. Conclusions are probably based on previous hypotheses or the results and conclusions of other studies, included in the review or included in the text, but are not frequently obtained through the results shown in the tables.
RESPONSE: Thank you for your positive comments and your helpful critique around the validity and reliability of the chosen literature. We appreciate this feedback, and it will be considered for future research, however, we would like to highlight that there is limited research within the UK/US contexts that specifically quantifiably answer the questions for this review. This review is contributing to the limited literature by synthesising the psycho-socioeconomic barriers to green space access for racialised individuals/families and BIPOC. Using a narrative synthesis methodology allows us to illuminate and apply key theories and ideas that have been addressed in other studies and reviews (such as the EJF [section 1.5] and issues around gentrification [section 1.7] – highlighted in the introduction section). We also acknowledge the need for more empirical research is needed to explore psychosocial and economic barriers to green space access for BIPOC and racialised individuals and families (line 605-607). This narrative review hopefully will act as a stimulus for such research.
Reviewer 4 Report
The paper overall is a worthwhile contribution to literature focused on unequal access to green spaces by racialized populations. Since the authors primarily investigated studies that took place in the UK and US, I feel that the Introduction is overly dependent on British studies to the near exclusion of US ones. The authors state (lines 498- 500) “our focus on a wide range of racialised groups (African, Caribbean, “Black”, Hispanic/Latino, west/south/south-east Asian), encompasses many racialised individuals/families in the UK and US offering robust ecological validity and generalisability.” To justify this statement, I recommend that the authors expand their citations to include such studies as: Ibes, Rakow, and Kim (2021), Rigolon and Nemeth, (2018), Plumer and Popovich, (2020) and other studies to expand the applicability.
--Barriers to Nature Engagement for Youth of Color - Ibes, Rakow, and Kim (2021)
-- Green gentrification or ‘just green enough’: Do park location, size and function affect whether a place gentrifies or not? - Rigolon and Nemeth, (2018)
-- How Racist Urban Planning Left Some Neighborhoods to Swelter - Plumer and Popovich, (2020)
Additional recommendations:
Line #
37 The term BIPOC is used most commonly in the US.
57-58 It is important to not imply that social prescribing can alleviate the inequities in the health system.
81-82 Sentence is redundant with previous sentence.
89-90 Important to state “within the UK.”
11-112 Here is where I would recommend additional papers representing the US and UK.
183 Table 2a- put into Appendices
449 “Findings suggest policy makers could initiate diversity-friendly schemes to improve engagement in green spaces for racialized communities.” Provide more exs. of how this could be done.
Author Response
FEEDBACK: The paper overall is a worthwhile contribution to literature focused on unequal access to green spaces by racialized populations. Since the authors primarily investigated studies that took place in the UK and US, I feel that the Introduction is overly dependent on British studies to the near exclusion of US ones. The authors state (lines 498- 500) “our focus on a wide range of racialised groups (African, Caribbean, “Black”, Hispanic/Latino, west/south/south-east Asian), encompasses many racialised individuals/families in the UK and US offering robust ecological validity and generalisability.” To justify this statement, I recommend that the authors expand their citations to include such studies as: Ibes, Rakow, and Kim (2021), Rigolon and Nemeth, (2018), Plumer and Popovich, (2020) and other studies to expand the applicability.
--Barriers to Nature Engagement for Youth of Color - Ibes, Rakow, and Kim (2021)
-- Green gentrification or ‘just green enough’: Do park location, size and function affect whether a place gentrifies or not? - Rigolon and Nemeth, (2018)
-- How Racist Urban Planning Left Some Neighborhoods to Swelter - Plumer and Popovich, (2020)
RESPONSE Thank you much for your and helpful feedback, highlighting research to further emphasise findings. Amendments made in the following sections:
- US research that was added into the review include:
- --Barriers to Nature Engagement for Youth of Color - Ibes, Rakow, and Kim (2021) – included in line 116-119 (reference 36).
- Green gentrification or ‘just green enough’: Do park location, size and function affect whether a place gentrifies or not? - Rigolon and Nemeth, (2018) – included in line 170 (reference 54)
- How Racist Urban Planning Left Some Neighborhoods to Swelter - Plumer and Popovich, (2020) – included in line 178 (reference 60).
We hope the inclusion of proposed studies adds to the US evidence base for this review.
- 37 The term BIPOC is used most commonly in the US. – We have added in the addition of BIPOC and its meaning in line 42 – 50. We have also used the term BIPOC throughout when referring to US studies.
- 57-58 It is important to not imply that social prescribing can alleviate the inequities in the health system. – We acknowledge this and have added in line 68 that social prescribing “can be one, of many, interventions used to improve their health and well-being and can be used alongside usual treatment”.
- 81-82 Sentence is redundant with previous sentence. – This line has been removed.
- 89-90 Important to state “within the UK.” – Please find in line 101 that this has been added.
- 11-112 Here is where I would recommend additional papers representing the US and UK. – Please find the studies have been added to in line 116-119, line 170 and line 178
- 183 Table 2a- put into Appendices – Please see that the appendices has been added in including tables A,B,C,D.
- 449 “Findings suggest policy makers could initiate diversity-friendly schemes to improve engagement in green spaces for racialized communities.” Provide more exs. of how this could be done. – Please see this has been added in line 510 – 512.
Round 2
Reviewer 3 Report
I have read your rewritten manuscript -systematic review- entitled: ”Examining psychosocial and economic barriers to green space access for racialized individuals and families”, ijerph-1984802, sent to the Journal of Environmental Research and Public Health.
Although I think that you have done your best to improve it. I consider that the methodology followed and the results present many and serious limitations to be published in a journal of the quality to the Journal of Environmental Research and Public Health.
Author Response
Thank you so much for your feedback. Although we appreciate that you are of the view that the methodology followed and the results present many and serious limitations, the methodology rigorously followed the process of a systematic review with a narrative synthesis (references 78 – Popay et al., [2006], and 79 – Campbell et al., [2020]). The benefits of this methodology for such research are that it can help to identify key research and any gaps (which in this case has real-world applications within US and UK contexts). This review has also been one of the first of its kind to further understand the barriers to green space access for racialised individuals/families; building on extensively researched theories, such as the Environmental Justice Framework.
Additionally, I would like to bring your attention to, similar systematic literature reviews that are published in the Journal of Environmental Research and Public Health. The following studies include similar limitations that have been highlighted as feedback for this review. Many of them use small samples and only reference one study for many points in their analysis. See below.
Geneshka, M., Coventry, P., Cruz, J., & Gilbody, S. (2021). Relationship between Green and Blue Spaces with Mental and Physical Health: A Systematic Review of Longitudinal Observational Studies. International journal of environmental research and public health, 18(17), 9010.
Lynch, M., Spencer, L. H., & Tudor Edwards, R. (2020). A systematic review exploring the economic valuation of accessing and using green and blue spaces to improve public health. International journal of environmental research and public health, 17(11), 4142.
MacMillan, F., George, E. S., Feng, X., Merom, D., Bennie, A., Cook, A., ... & Astell-Burt, T. (2018). Do natural experiments of changes in neighborhood built environment impact physical activity and diet? A systematic review. International journal of environmental research and public health, 15(2), 217.
Equally, the study by MacMillan et al., 2018 highlighted “However, study design (lack of a comparison group), underpowered sample sizes, the use of a wide array of outcome measures and limited reporting in some included studies, have made it challenging to draw overall conclusions in this review.” Despite the limitations highlighted, this journal has still published the research.
We hope that with this in mind, you can review your comments and consider this manuscript for publication, as we acknowledge the limitations of this review, but we believe the findings of this review are worth publishing, as they illuminate racialised individuals/families experiences of green space access that can add to the limited existing literature.